# Vitamin B12 deficiency in diabetic patients treated with metformin: A cross-sectional study

**Dat Tan Huynh[1], Ngoc Thi Nguyen[2], Minh Duc Do** [3]*

**1** Department of Endocrinology, Faculty of Medicine, University of Medicine and Pharmacy at Ho Chi Minh City, Ho Chi Minh City, Vietnam, **2** Vimec Central Park Hospital, Ho Chi Minh City, Vietnam, **3** Center for Molecular Biomedicine, University of Medicine and Pharmacy at Ho Chi Minh City, Ho Chi Minh City, Vietnam

* ducminh@ump.edu.vn

**Data Availability Statement:** All relevant data are within the paper and its Supporting Information files.

**Funding:** The author(s) received no specific funding for this work.

## Abstract

Metformin is a cornerstone medication in the management of type 2 diabetes. Metformin is safe, effective, and inexpensive; however, it is associated with vitamin B12 deficiency. This study set out to evaluate the rate of vitamin B12 deficiency in Vietnamese patients with type 2 diabetes who were treated with metformin and to investigate factors associated with vitamin B12 deficiency. This is a cross-sectional study that was conducted in Vinmec Central Park Hospital from February to June 2023. The clinical and paraclinical characteristics of the participants were recorded, and the levels of vitamin B12 and folate were measured. The rate of vitamin B12 deficiency in patients treated with metformin was found to be 18.6%. Further, the duration of diabetes, duration of metformin use, metformin dose, and hemoglobin levels were statistically associated with vitamin B12 deficiency with OR (95% CI) = 1.12 (1.03–1.19), 1.01 (1.00–1.02), 1.002 (1.001–1.002), 0.74 (0.55–0.99), respectively. After adjusting for covariates, a metformin dose greater than the median dose remained the only parameter associated with vitamin B12 deficiency, with OR (95% CI) = 4.10 (1.62–10.36). Moreover, when combining both long-term use of metformin and a metformin dose greater than the median dose, the OR increased to 5.25 (95% CI: 2.11–13.15). These results demonstrate that vitamin B12 deficiency in patients treated with metformin is quite prevalent in Vietnam and that those with long-term use of metformin (48 months or more) and high metformin dose (1000 mg/day or more) are at high risk of experiencing this adverse effect and so require screening.

## Introduction

Type 2 diabetes mellitus (T2DM) is one of the most prevalent and burdensome non-communicable diseases globally, and the pathophysiology of the disease comprises both environmental and genetic factors [1–4]. The aim of treatment of T2DM is to control blood glucose levels in the target and so avoid disease complications affecting the patient's quality of life and life expectancy [5]. Since its discovery in 1922, metformin has become one of the most prescribed

**Competing interests:** The authors have declared that no competing interests exist.

oral medications in the management of T2DM. The mechanisms of action of metformin include reducing glucose neogenesis in the liver and increasing insulin sensitivity in peripheral tissues, such as muscle and the liver [6]. Thus, metformin is very useful for glycemic control, and it helps reduce long-term complications of T2DM [7]. Although metformin is cheap, effective, and widely used, it has certain side effects that can decrease drug tolerance. One of the most neglected side effects of metformin is vitamin B12 deficiency. The precise mechanisms for metformin-induced vitamin B12 deficiency are still not fully understood; however, several potential mechanisms have been proposed, including reduced intrinsic factor secretion, inhibition of intrinsic factor-vitamin B12 complex absorption, changes in bile acid metabolism and reabsorption, and interference in the binding of the intrinsic factor-vitamin B12 complex to the cubilin receptor [8,9]. Long-term vitamin B12 deficiency can have severe consequences, such as anemia, hearing loss, and neuropathy (which is frequently misdiagnosed as diabetic peripheral neuropathy) [10,11].

The first report of metformin-induced vitamin B12 deficiency was published in 1971 [12]. After that, the HOME randomized control trial has undoubtedly confirmed the causative role of metformin in vitamin B12 deficiency [13]. Several meta-analysis studies also showed a strong association between metformin use and vitamin B12 deficiency in diabetic patients [14–17]. Current guidelines recommend annual measurement of vitamin B12 in patients undergoing long-term treatment with metformin [18]. However, the prevalence of metformin-induced vitamin B12 deficiency differs between locations and ethnicities; the rate varies from 4.3% up to 30% [12,19–21]. In Vietnam, there were around 4,000,000 diabetic patients in 2021 [3], and the majority of these patients were treated with metformin. Despite the common use of metformin, no assessment of the rate of vitamin B12 deficiency in Vietnam has been published. Therefore, this study set out to investigate the rate of vitamin B12 deficiency and associated factors in Vietnamese T2DM patients treated with metformin.

## Materials and methods

### Patient recruitment

This is a cross-sectional study that was conducted with outpatients in Vinmec Central Park Hospital from February 2023 to June 2023. The protocol of this study was approved by the Ethical Committee of University of Medicine and Pharmacy at Ho Chi Minh City (approval number 1044/HDDD-DHYD). The patients gave their written informed consents to participate in this study. The inclusion criteria for recruitment were: patients diagnosed with T2DM who had been treated with metformin for at least 6 months and whose dose had remained unchanged in the last 6 months.

The exclusion criteria were: infection, cancer, heart failure, pernicious anemia, liver enzymes aspartate aminotransferase (AST) and alanine transaminase (ALT) more than three times the upper limit, and estimated glomerular filtration rate less than 30 ml/minute/1.73 m$^2$ body surface area. We also excluded patients who had a history of gastrointestinal issues (such as gastro-duodenal resection, pancreatectomy, chronic colitis, or irritable bowel syndrome) or had used a proton pump inhibitor or $H_2$ receptor antagonist continuously in the previous 12 months. Patients with a wholly vegan diet, a history of alcoholic abuse ($\geq$ 3 alcoholic units per day), who took vitamin B12/multivitamin supplements, or who were pregnant were also excluded from the study.

The sample size of the study was estimated using the following formula:

$$N = Z^2_{\left(1-\frac{\alpha}{2}\right)} \frac{p(1-p)}{d^2}$$

With Z = 1.96, p = 0.22 (based on a similar study in Korea in 2019), and d = 0.065, the estimated patient number was 156. After patients agreed to participate in the study by providing written informed consent, clinical information was collected by the investigators. Baseline characteristics were documented, including age, sex, height, weight, blood pressure, duration of T2DM, duration of metformin use, and metformin dose. Patients were considered long-term users of metformin when they had been treated with metformin for 48 consecutive months or longer.

### Laboratory measurements

Peripheral blood samples were collected from participants and sent to the central laboratory of Vinmec Central Park Hospital. Vitamin B12 and acid folic levels were measured using a Uni-Cel DxI 800 Access Immunoassay System (Beckman Coulter, CA, USA). Total blood count was obtained using a DxH 600 Hematology Analyzer (Beckman Coulter), and fasting plasma glucose, creatinine, AST, ALT, and lipid profile were measured using an AU680 Clinical Chemistry Analyzer (Beckman Coulter). HbA1c was measured by a Premier Hb9210 HbA1c Analyzer (Trinity Biotech, Ireland). Vitamin B12 deficiency was defined as a vitamin B12 level < 300pg/ml with a normal level of folic acid ($\geq$ 4ng/ml).

### Statistical analysis

Frequencies and percentages were used to express binominal variables. The independence of binominal variables was evaluated using Chi-squared or Fisher's exact tests. The normal distribution of continuous variables was checked by the Kolmogorov-Smirnov test; these variables were presented as either mean with standard deviation (SD) if they were normally distributed or median with interquartile range [IQR] if they were not normally distributed. The differences between continuous variables were tested by Student's t-test or the Mann-Whitney U test based on the data distribution. Factors associated with vitamin B12 deficiency were assessed by odds ratio (OR) and 95% CI in univariate and multivariable logistic regression analyses. Statistical analysis was performed using SPSS Statistics for Windows version 20.0 (IBM Corp., Armonk, NY, USA). A p-value of less than 0.05 was considered to be statistically significant.

## Results

### Baseline characteristics of participants

The characteristics of the participants are described in Table 1. The mean age of the studied population was 56.3, the median diabetes duration was 4.8 years, the median duration of metformin use was 48 months, and the median dose of metformin was 1000mg. Overall, the rate of vitamin B12 deficiency was 18.6% (29 out of 156 cases). Folate deficiency and macrocytic anemia were not found in any of the participants. Patients with vitamin B12 deficiency had significantly longer duration of diabetes, longer duration of metformin use, higher metformin dose, lower cholesterol levels, and lower LDL-C levels.

### Factors associated with vitamin B12 deficiency

Univariate logistic regression analysis showed that factors statistically associated with vitamin B12 deficiency were duration of diabetes, duration of metformin use, metformin dose, and hemoglobin level (Table 2). Long-term use of metformin and metformin dose > median was statistically associated with vitamin B12 deficiency, and ORs were 3.91 and 4.16, respectively.

**Table 1. Baseline characteristics of the participants.**

| Characteristics | Total (N = 156) | No vitamin B12 deficiency (N = 127) | Vitamin B12 deficiency (N = 29) | p-value |
|---|---|---|---|---|
| Age (years)<br>Mean ± SD | 56.3 ± 12.3 | 55.5 ± 11.8 | 59.8 ± 13.9 | 0.13 |
| Gender<br>  Female, N (%)<br>  Male, N (%) | <br>69 (44.2)<br>87 (55.8) | <br>57 (44.9)<br>70 (55.1) | <br>12 (41.4)<br>17 (58.6) | 0.45 |
| BMI (kg/m$^2$)<br>Mean ± SD | 24.6 ± 3.4 | 24.7 ± 3.6 | 24.1 ± 2.1 | 0.29 |
| SBP (mmHg)<br>Mean ± SD | 127.9 ± 15.1 | 127.6 ± 15.2 | 128.8 ± 14.9 | 0.72 |
| DBP (mmHg)<br>Mean ± SD | 72.9 ± 9.8 | 73.4 ± 9.7 | 71.2 ± 10.0 | 0.30 |
| Diabetes duration (years)<br>Median [Q1-Q3] | 4.8 [2.0–8.3] | 3.5 [1.3–6.3] | 6.0 [5.0–10.0] | < 0.01 |
| Metformin use duration (months)<br>Median [Q1-Q3] | 48.0 [13.5–84.0] | 36.0 [12.0–60.0] | 60.0 [48.0–120.0] | < 0.01 |
| Metformin dose (mg)<br>Median [Q1-Q3] | 1000.0 [1000.0–1500.0] | 1000.0 [1000.0–1000.0] | 1500.0 [1000.0–2000.0] | < 0.01 |
| Hb (g/dl)<br>Mean ± SD | 14.3 ± 1.4 | 14.4 ± 1.3 | 13.8 ± 1.6 | 0.07 |
| MCV (fL)<br>Mean ± SD | 87.7 ± 6.2 | 87.9 ± 5.9 | 86.8 ± 1.6 | 0.44 |
| Fasting plasma glucose (mmol/l)<br>Mean ± SD | 7.8 ± 2.7 | 7.7 ± 2.8 | 8.0 ± 2.2 | 0.52 |
| HbA1c (%)<br>Mean ± SD | 7.6 ± 1.7 | 7.5 ± 1.7 | 8.0 ± 1.5 | 0.15 |
| Creatinine (mg/dl)<br>Mean ± SD | 78.2 ± 16.9 | 77.1 ± 16.1 | 82.9 ± 19.2 | 0.15 |
| AST (U/L)<br>Median [Q1-Q3] | 27.7 [21.5–35.9] | 27.8 [21.9–37.0] | 26.0 [19.7–33.9] | 0.42 |
| ALT (U/L)<br>Median [Q1-Q3] | 26.1 [19.4–38.1] | 26.1 [19.6–38.1] | 24.1 [19.2–35.4] | 0.96 |
| Total cholesterol (mmol/l)<br>Median [Q1-Q3] | 4.5 [3.6–5.5] | 4.6 [3.7–5.6] | 4.2 [3.2–4.6] | 0.02 |
| Triglyceride (mmol/l)<br>Median [Q1-Q3] | 1.6 [1.2–2.4] | 1.6 [1.2–2.4] | 1.8 [1.1–2.3] | 0.95 |
| HDL-C (mmol/l)<br>Median [Q1-Q3] | 1.2 [1.0–1.4] | 1.2 [1.1–1.4] | 1.1 [0.9–1.3] | 0.79 |
| LDL-C (mmol/l)<br>Median [Q1-Q3] | 2.8 [2.1–3.5] | 2.8 [2.2–3.5] | 2.3 [1.8–2.9] | 0.02 |
| Vitamin B12 level (pg/ml)<br>Median [Q1-Q3] | 450.0 [335.3–594.0] | 483.0 [403.0–613.0] | 255.0 [231.0–280.0] | < 0.01 |
| Folate level (μmol/mL)<br>Mean ± SD | 13.0 ± 4.0 | 12.8 ± 3.9 | 13.8 ± 4.6 | 0.31 |

BMI: body mass index, SBP: systolic blood pressure, DBP: diastolic blood pressure, Hb: hemoglobin, MCV: mean cell volume, HDL-C: high-density lipoprotein-cholesterol, LDL-C: low-density lipoprotein-cholesterol.

When combining both long-term use of metformin and metformin dose > median dose, the OR increased to 5.25 (95% CI: 2.11–13.15) (Fig 1).

After adjusting for covariates using the two models (each model contained four variables) shown in Table 3, it was found that only metformin dose was associated with vitamin B12 deficiency in both models. Each 1mg increase in metformin dose was associated with a 0.2%

**Table 2. Univariate logistic regression analysis of factors associated with vitamin B12 deficiency.**

| Variables | OR (95% CI) | p-value |
|---|---|---|
| Age | 1.03 (0.99–1.07) | 0.91 |
| Gender | 0.87 (0.38–1.96) | 0.73 |
| BMI | 0.95 (0.84–1.08) | 0.44 |
| Duration of diabetes | 1.12 (1.03–1.19) | < 0.01 |
| Duration of metformin use | 1.01 (1.00–1.02) | 0.05 |
| Long-term use of metformin | 3.91 (1.61–9.51) | < 0.01 |
| Metformin dose | 1.002 (1.001–1.002) | < 0.01 |
| Metformin dose > median dose | 4.16 (1.78–9.70) | < 0.01 |
| Hb | 0.74 (0.55–0.99) | 0.04 |
| MCV | 0.97 (0.92–1.03) | 0.37 |
| Fasting plasma glucose | 1.04 (0.90–1.20) | 0.58 |
| HbA1c | 1.17 (0.93–1.46) | 0.18 |
| Folate level | 1.06 (0.96–1.17) | 0.25 |

BMI: body mass index, Hb: hemoglobin, MCV: mean cell volume.

increase in the risk of vitamin B12 deficiency. Metformin dose > median dose (>1000mg daily) was independently associated with vitamin B12 deficiency with OR = 4.10 (95% CI: 1.62–10.36).

## Discussion

The rate of vitamin B12 deficiency in T2DM patients treated with metformin was found in this study to be 18.6%. This result is similar to the results from a Korean study using the same criterion for vitamin B12 deficiency, namely a level of less than 300 pg/mL [22]. The rate, however, was up to 29.1% in a study conducted in the United States in which the criterion for vitamin B12 deficiency was a level of less than 350pg/mL [23]. On the other hand, the rate of vitamin B12 deficiency in patients treated with metformin was only 14.1% in a study in the Netherlands in which the criterion was a level of less than 150 pmol/L [24]. These differences might

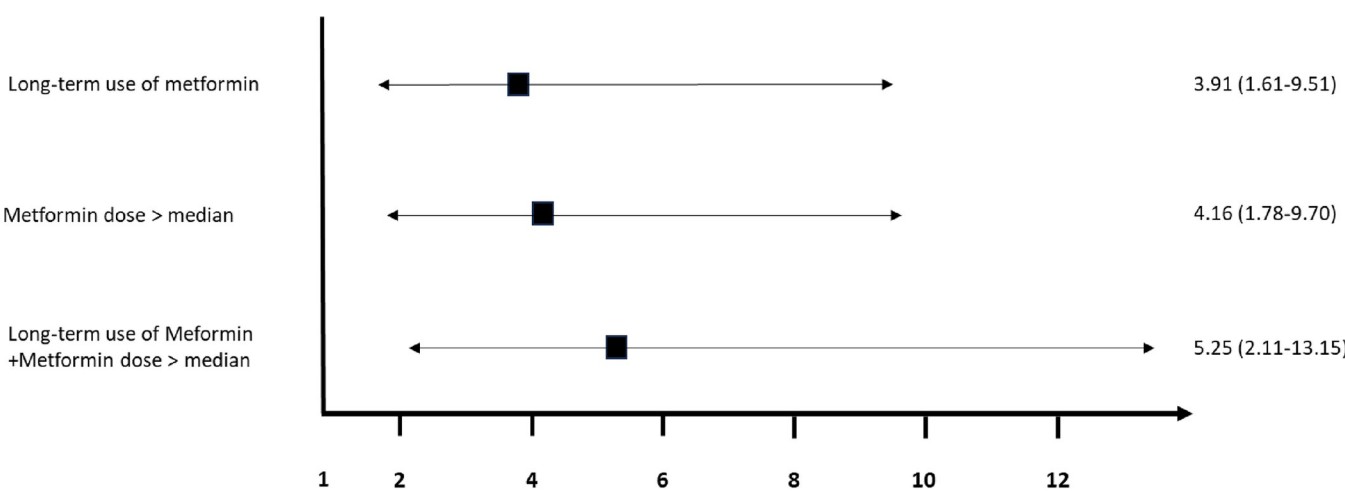

**Fig 1. Odd ratios of long-term use of metformin and metformin dose > median for vitamin B12 deficiency.**

**Table 3. Multivariate logistic regression analysis of factors associated with vitamin B12 deficiency.**

| Model | Variables | OR (95% CI) | p-value |
|---|---|---|---|
| Model 1 | Diabetes duration | 1.07 (0.91–1.26) | 0.39 |
| | Duration of metformin use | 1.01 (0.85–1.20) | 0.88 |
| | Metformin dose | 1.002 (1.001–1.003) | < 0.01 |
| | Hemoglobin | 0.71 (0.51–0.99) | 0.04 |
| Model 2 | Diabetes duration | 1.05 (0.96–1.14) | 0.27 |
| | Long-term use of metformin | 2.24 (0.69–7.30) | 0.18 |
| | Metformin dose > median dose | 4.10 (1.62–10.36) | < 0.01 |
| | Hemoglobin | 0.77 (0.56–1.07) | 0.11 |

be explained by differences in measuring methods, ethnicities, or diet. However, in general, the more stringent the criteria used, the lower the detection rate for vitamin B12 deficiency.

Patients with vitamin B12 deficiency had a significantly longer duration of diabetes and longer exposure to metformin, as well as higher daily doses of metformin. These variables were also shown to be statistically associated with vitamin B12 deficiency in univariate logistic regression analysis. The observation that vitamin B12 deficiency is associated with duration of diabetes is similar to the findings of studies conducted in Yongin Severance Hospital, the Madigan Army Medical Center Family Medicine Clinic, and the Diabetic and Endocrine Centre at Al-Noor Specialist Hospital [22,23,25]; however, this association was not found in a study in Karachi, Pakistan [26]. The duration of metformin use and metformin dose are two important factors that have been reported to be associated with vitamin B12 deficiency in several meta-analyses [14,16]. However, these two factors are not always associated with vitamin B12 deficiency, as shown in a study in Saudi Arabia [25]. These inconsistencies can be explained by the fact that there are differences in the inclusion criteria, regions, vitamin supplements, and composition of diet. In this study, the long-term use of metformin (48 months or more) was strongly associated with vitamin B12 deficiency in the univariate analysis, with OR = 3.91. This finding together with recent real-world evidence [27] once again suggests that patients on long-term metformin treatment should be screened for vitamin B12 deficiency.

In the multivariate logistic regression analysis, metformin dose was the only parameter associated with vitamin B12 deficiency in both models. This result emphasizes the important role of metformin dose, not just duration of metformin use, in vitamin B12 deficiency. Despite choosing a low dose threshold of 1000mg, we observed a significant association of high metformin dose (> 1000mg daily) with vitamin B12 deficiency. Several studies have shown similar results, with metformin dose a strong factor associated with vitamin B12 deficiency [17,24,28,29]; thus, screening for this side effect needs to be performed in patients with either long-term use or a high metformin dose.

This study has limitations that need to be discussed. First, methylmalonic acid and homocysteine levels were not measured, and this may lead to the underdiagnosis of early vitamin B12 deficiency. In cellular level, vitamin B12 helps converting homocysteine to methionine, therefore, vitamin B12 deficiency will lead to increased homocysteine level [30]. Furthermore, vitamin B12 promotes the conversion of methylmalonyl-CoA to succinyl-CoA in mitochondria. Vitamin B12 deficiency results in the accumulation of methylmalonyl-CoA which will later convert to methylmalonic acid [30]. The increase of both methylmalonic acid and homocysteine levels reflects vitamin B12 deficiency in tissues and can cause of worsen diabetic peripheral neuropathy [31,32]. Second, participants were not systematically assessed for peripheral neuropathy, an important consequence of vitamin B12 deficiency. Due to the lack of resources, we could not measure methylmalonic acid and homocysteine levels as well as comprehensively examine diabetic peripheral neuropathy. Third, due to the nature of the

cross-sectional study, we could not ascertain the causative effect of metformin on vitamin B12 deficiency. Finally, participants were only recruited in a single center, and the rate of vitamin B12 deficiency reported here may not represent the wider population of Vietnamese diabetic patients treated with metformin.

To the best of our knowledge, this is the first study in Vietnam investigating vitamin B12 deficiency in type 2 diabetes patients treated with metformin. We found that 18.6% of patients treated with metformin for at least 6 months had vitamin B12 deficiency. This result will pave the way for further studies with larger sample sizes, multi-center recruitment and, ideally, a longitudinal design to improve understanding of the role of metformin in vitamin B12 deficiency.

## Supporting information

**S1 Data. Data of the study.**
(XLSX)

## Author Contributions

**Conceptualization:** Dat Tan Huynh, Minh Duc Do.

**Data curation:** Dat Tan Huynh, Ngoc Thi Nguyen, Minh Duc Do.

**Formal analysis:** Dat Tan Huynh, Minh Duc Do.

**Investigation:** Dat Tan Huynh, Ngoc Thi Nguyen, Minh Duc Do.

**Writing – original draft:** Dat Tan Huynh, Ngoc Thi Nguyen, Minh Duc Do.

**Writing – review & editing:** Minh Duc Do.

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
