## [Editor Report · Decision Letter 0]

18 Jan 2024

PONE-D-24-00863VITAMIN B12 DEFICIENCY IN DIABETIC PATIENTS TREATED WITH METFORMIN: A CROSS-SECTIONAL STUDYPLOS ONE

Dear Dr. Do,

Thank you for submitting your manuscript to PLOS ONE. After careful consideration, we feel that it has merit but does not fully meet PLOS ONE’s publication criteria as it currently stands. Therefore, we invite you to submit a revised version of the manuscript that addresses the points raised during the review process.

We look forward to receiving your revised manuscript.

Kind regards,

Joseph Alan Bauer, Ph.D.

Academic Editor

PLOS ONE

Journal Requirements:

5. Please include a caption for figure 1.

Additional Editor Comments (if provided):

The manuscript titled "VITAMIN B12 DEFICIENCY IN DIABETIC PATIENTS TREATED WITH METFORMIN: A CROSS-SECTIONAL STUDY" addresses a significant issue in diabetic care. The relationship between long-term metformin use and vitamin B12 deficiency is well-documented in the literature, with several studies indicating a clear association. This is an area of concern given the widespread use of metformin in managing type 2 diabetes.

Your manuscript should consider incorporating findings from various studies that have explored this relationship. For instance, the HOME trial, as published in the BMJ in 2010, provides valuable insights into the effects of metformin on vitamin B12 levels in diabetic patients. This randomized placebo-controlled trial observed changes in vitamin B12 levels over different phases of metformin treatment, offering a structured and comprehensive understanding of the issue.

Further, a systematic review on PubMed highlights various articles that delve into the association between metformin use, vitamin B12 deficiency, and its clinical implications like peripheral neuropathy. These studies collectively underscore the importance of monitoring vitamin B12 levels in diabetic patients on long-term metformin therapy.

Additionally, the association between metformin-induced vitamin B12 deficiency and peripheral neuropathy, as discussed in several articles on PubMed, is a critical aspect that should be addressed in your manuscript. The prevalence of vitamin B12 deficiency among diabetic patients on metformin and its potential impact on peripheral neuropathy is a significant area of concern, and your study could benefit from discussing these findings.

Regarding the use of homocysteine (HCY) and methylmalonic acid (MMA) as indicators of vitamin B12 deficiency in tissues, these are reliable markers for diagnosing vitamin B12 deficiency at the tissue level. Their omission in the experimental design of your study is a point that should be addressed. The rationale for not including these markers should be clarified, considering their relevance in understanding the biochemical and clinical implications of vitamin B12 deficiency.

It is also advisable to incorporate open-access references, as they would align with the broader accessibility goals of current scientific research. The inclusion of such references would not only enhance the credibility of your manuscript but also ensure its wider reach and impact.

Finally, conducting a current literature search through databases like PubMed would provide the most recent and relevant studies, ensuring that your manuscript reflects the current understanding and research trends in this field. https://pubmed.ncbi.nlm.nih.gov/?term=metformin+b12+deficiency&sort=date

---

## [Author Response · Author response to Decision Letter 0]

22 Feb 2024

Response to Reviewers:

The manuscript titled "VITAMIN B12 DEFICIENCY IN DIABETIC PATIENTS TREATED WITH METFORMIN: A CROSS-SECTIONAL STUDY" addresses a significant issue in diabetic care. The relationship between long-term metformin use and vitamin B12 deficiency is well-documented in the literature, with several studies indicating a clear association. This is an area of concern given the widespread use of metformin in managing type 2 diabetes.

Your manuscript should consider incorporating findings from various studies that have explored this relationship. For instance, the HOME trial, as published in the BMJ in 2010, provides valuable insights into the effects of metformin on vitamin B12 levels in diabetic patients. This randomized placebo-controlled trial observed changes in vitamin B12 levels over different phases of metformin treatment, offering a structured and comprehensive understanding of the issue.

Thank you so much for your suggestions, we’ve added this information to the Introduction part to emphasize the important role of the HOME trial as one of the randomized control trials confirming the causative role of metformin in vitamin B12 deficiency[1].

Further, a systematic review on PubMed highlights various articles that delve into the association between metformin use, vitamin B12 deficiency, and its clinical implications like peripheral neuropathy. These studies collectively underscore the importance of monitoring vitamin B12 levels in diabetic patients on long-term metformin therapy.

Thank you for your suggestions, we’ve also added some of the systematic reviews in the Introduction and Discussion part[2–4].

Additionally, the association between metformin-induced vitamin B12 deficiency and peripheral neuropathy, as discussed in several articles on PubMed, is a critical aspect that should be addressed in your manuscript. The prevalence of vitamin B12 deficiency among diabetic patients on metformin and its potential impact on peripheral neuropathy is a significant area of concern, and your study could benefit from discussing these findings.

Thank you so much for pointing out this weakness in our study. The peripheral neuropathy complication is often observed in diabetic patients with vitamin B12 deficiency. However, due to the lack of resources (funding and the participation of a neurologist), we could not make a definitive diagnosis of peripheral neuropathy in the studied population. This is obviously a major drawback of this study, and we also admit that in the Discussion part.

Regarding the use of homocysteine (HCY) and methylmalonic acid (MMA) as indicators of vitamin B12 deficiency in tissues, these are reliable markers for diagnosing vitamin B12 deficiency at the tissue level. Their omission in the experimental design of your study is a point that should be addressed. The rationale for not including these markers should be clarified, considering their relevance in understanding the biochemical and clinical implications of vitamin B12 deficiency.

Thank you for your suggestions, we’ve also added the information regarding the importance of homocysteine (HCY) and methylmalonic acid (MMA) as indicators of vitamin B12 deficiency in tissues in Discussion part, we also acknowledge that the increase of both methylmalonic acid and homocysteine levels can cause of worsen diabetic peripheral neuropathy[5,6]. As we previously stated, due to the lack of resources, we could not measure these important markers.

It is also advisable to incorporate open-access references, as they would align with the broader accessibility goals of current scientific research. The inclusion of such references would not only enhance the credibility of your manuscript but also ensure its wider reach and impact. Finally, conducting a current literature search through databases like PubMed would provide the most recent and relevant studies, ensuring that your manuscript reflects the current understanding and research trends in this field. https://pubmed.ncbi.nlm.nih.gov/?term=metformin+b12+deficiency&sort=date

Thank you for your suggestions, we’ve added more open access references in the recent years on Pubmed[4,7,8], we hope that this information could reflect the current understanding and research trends in this field.

References:

1. de Jager J, Kooy A, Lehert P, Wulffelé MG, van der Kolk J, Bets D, et al. Long term treatment with metformin in patients with type 2 diabetes and risk of vitamin B-12 deficiency: randomised placebo controlled trial. The BMJ. 2010;340: c2181. doi:10.1136/bmj.c2181

2. Khattab R, Albannawi M, Alhajjmohammed D, Alkubaish Z, Althani R, Altheeb L, et al. Metformin-Induced Vitamin B12 Deficiency among Type 2 Diabetes Mellitus’ Patients: A Systematic Review. Curr Diabetes Rev. 2023;19: e180422203716. doi:10.2174/1573399818666220418080959

3. Chapman LE, Darling AL, Brown JE. Association between metformin and vitamin B12 deficiency in patients with type 2 diabetes: A systematic review and meta-analysis. Diabetes Metab. 2016;42: 316–327. doi:10.1016/j.diabet.2016.03.008

4. Kakarlapudi Y, Kondabolu SK, Tehseen Z, Khemani V, J SK, Nousherwani MD, et al. Effect of Metformin on Vitamin B12 Deficiency in Patients With Type 2 Diabetes Mellitus and Factors Associated With It: A Meta-Analysis. Cureus. 2022;14. doi:10.7759/cureus.32277

5. Wile DJ, Toth C. Association of Metformin, Elevated Homocysteine, and Methylmalonic Acid Levels and Clinically Worsened Diabetic Peripheral Neuropathy. Diabetes Care. 2009;33: 156–161. doi:10.2337/dc09-0606

6. Bell DSH. Metformin-induced vitamin B12 deficiency can cause or worsen distal symmetrical, autonomic and cardiac neuropathy in the patient with diabetes. Diabetes Obes Metab. 2022;24: 1423–1428. doi:10.1111/dom.14734

7. Liu Q, Li S, Quan H, Li J. Vitamin B12 status in metformin treated patients: systematic review. PloS One. 2014;9: e100379. doi:10.1371/journal.pone.0100379

8. Hurley-Kim K, Vu CH, Dao NM, Tran LC, McBane S, Lee J, et al. Effect of Metformin Use on Vitamin B12 Deficiency Over Time (EMBER): A Real-World Evidence Database Study. Endocr Pract. 2023;29: 862–867. doi:10.1016/j.eprac.2023.06.013

---

## [Editor Report · Decision Letter 1]

5 Apr 2024

VITAMIN B12 DEFICIENCY IN DIABETIC PATIENTS TREATED WITH METFORMIN: A CROSS-SECTIONAL STUDY

PONE-D-24-00863R1

Dear Dr. Do,

We’re pleased to inform you that your manuscript has been judged scientifically suitable for publication and will be formally accepted for publication once it meets all outstanding technical requirements.

Kind regards,

Joseph Alan Bauer, Ph.D.

Academic Editor

PLOS ONE